# A Simplified and Efficient Method for Production of Manganese Ferrite Magnetic Nanoparticles and Their Application in DNA Isolation

**DOI:** 10.3390/ijms24032156

**Published:** 2023-01-21

**Authors:** Tímea B. Gerzsenyi, Ágnes M. Ilosvai, Gergely Szilágyi, Milán Szőri, Csaba Váradi, Béla Viskolcz, László Vanyorek, Emma Szőri-Dorogházi

**Affiliations:** 1Higher Education and Industrial Cooperation Centre, University of Miskolc, 3515 Miskolc, Hungary; 2Institute of Chemistry, Faculty of Materials and Chemical Engineering, University of Miskolc, 3515 Miskolc, Hungary

**Keywords:** magnetic phase, DNA isolation, ferrite, hydrogen bridge

## Abstract

A simplified, fast, and effective production method has been developed for the synthesis of manganese ferrite (MnFe_2_O_4_) magnetic nanoparticles (MNPs). In addition to the wide applicability of MnFe_2_O_4_ MNPs, this work also reports their application in DNA isolation for the first time. An ultrasonic-cavitation-assisted combustion method was applied in the synthesis of MnFe_2_O_4_ MNPs at different furnace temperatures (573 K, 623 K, 673 K, and 773 K) to optimize the particles’ properties. It was shown that MnFe_2_O_4_ nanoparticles synthesized at 573 K consist of a spinel phase only with adequate size and zeta potential distributions and superparamagnetic properties. It was also demonstrated that superparamagnetic manganese ferrite nanoparticles bind DNA in buffer with a high NaCl concentration (2.5 M), and the DNA desorbs from the MNPs by decreasing the NaCl concentration of the elution buffer. This resulted in a DNA yield comparable to that of commercial DNA extraction products. Both the DNA concentration measurements and electrophoresis confirmed that a high amount of isolated bacterial plasmid DNA (pDNA) with adequate purity can be extracted with MnFe_2_O_4_ (573 K) nanoparticles by applying the DNA extraction method proposed in this article.

## 1. Introduction

MnFe_2_O_4_ is one of the most promising spinel ferrite nanoparticles (NPs) [1], since it is useful in fuel production, can serve as an anode material for Li^+^ batteries, and can also be used in the removal of heavy metals as adsorbents for wastewater treatment [2], as well as being used as a catalyst [3]. MnFe_2_O_4_ NPs are also efficient candidates for various biomedical applications such as in drug delivery, magnetic resonance imaging (MRI) [4,5,6,7], or as a heat source in magnetic hyperthermia-mediated cancer therapy [8]. The main reason for the wide applicability of MnFe_2_O_4_ NPs is their ferrimagnetic property, which can be controlled by the nanoparticle size. If a magnetic field is applied, magnetic moments of the magnetic domains align with the magnetic field, which results in a large net magnetic moment [9,10,11,12]. When the nanoparticles’ size decreases below a threshold value (commonly 20 nm [13]), the ferrimagnetic material becomes a single domain that is characterized by a uniform magnetization, and these nanoparticles have a larger, localized magnetic field compared to those of larger particles [9,10].

By changing the synthesis method, operation conditions, and the concentration of precursors, MnFe_2_O_4_ NPs can be produced in different sizes, morphologies, purities, and crystallinities, thus NPs can be easily adjusted according to the need of the application [3]. There are various preparation methods for magnetic material synthesis such as solvothermal synthesis [14], classical co-precipitation [15], hydrothermal reactions [16], sol–gel synthesis [17], thermal decomposition [18], microemulsion synthesis [19], sonochemical reactions [20], electrospray synthesis [21], and laser pyrolysis [22].

In the last two decades, nanoparticle-based biotechnology has shown great progress and has become a promising field due to its numerous applications. For instance, magnetic beads can be used for the construction of a novel modular-chip-based bacterial DNA extraction devices [23], in the isolation/pre-concentration of various target molecules, or in biomedical applications such as biosensing platforms [24]. Recently, iron oxide magnetic nanoparticles pre-mixed with DNA molecules have appeared as a potential component of tissue engineering scaffolds, since MNP–DNA-modified surfaces promote differentiation of mesenchymal stem cells [25]. A new hybrid magnetic composite (iron oxide nanoparticles coated with different polymers) has been used in novel, practical, and efficient pDNA extraction and purification protocols as well [26]. pDNA is a double-stranded extra-chromosomal DNA molecule of a relatively small size, high stability, and autonomous replication capability, thus it is an indispensable tool for genetic engineering. pDNA is usually used as a starting material in most biotechnology procedures, therefore the development of new, efficient protocols for pDNA extraction from bacterial cells is a hot topic [27]. pDNA separation from cell lysate with magnetic nanoparticles has several advantages compared to time-consuming, conventional isolation with toxic organic solvents or to the cartridge-based isolation method which requires several centrifugation steps and expensive kits. The magnetic solid-phase extraction method speeds up nucleic acid isolation from crude samples such as cell lysates [28] and thus enables the downstream applications of molecular biology to commence faster, such as amplification, cloning, sequencing, or hybridization [29,30,31].

To the best of our knowledge, MnFe_2_O_4_ MNPs have never been tested in nucleic acid isolation. Thus, the aim of this work was to achieve simple, fast, and efficient synthesis of MnFe_2_O_4_ magnetic nanoparticles that can then be further tested in molecular biological applications.

To produce manganese ferrite nanoparticles, combustion, and sonochemical treatments can be combined. The sonochemical method can provide a high reaction rate and controllable synthesis conditions, yielding high purity NPs, thus allowing a narrow size distribution, scalability, and environmental friendliness [32,33,34,35]. To get rid of the remaining organic compounds from the surface of the nanoparticles, a combustion step can be conducted. The great advantage of the method is that it consists of two simple steps, and it is quick and easy to implement without of need for washing, filtering, or centrifugation like in traditional co-precipitation procedures [36,37,38].

After the sonochemical-assisted combustion synthesis of MnFe_2_O_4_ magnetic nanoparticles, the physico-chemical characterization of these MNPs was carried out by X-ray diffraction (XRD), transmission electron microscopy (TEM), Fourier transform infrared spectroscopy (FTIR), and magnetization hysteresis experiments. Thereafter the bacterial pDNA binding capacity of nanoparticles was tested.

## 2. Results and Discussion

### 2.1. XRD Characterization

The presence of magnetically separable manganese ferrite spinel in the samples was confirmed by XRD measurements. As Figure 1 shows, the XRD reflection peaks were located at 18.1° (111), 29.9° (220), 35.3° (311), 36.8° (222), 42.5° (400), 52.7° (422), 56.3° (511), and 61.7° (440) two theta degrees, which match with the peaks corresponding to the manganese ferrite phase in all of the samples that were produced at the following temperatures: 573 K, 623 K, 673 K, and 773 K (PDF 74-2403). These reflections were also characteristic for manganese ferrite samples created by the co-precipitation method [36,37,38]. 

By increasing the temperature to 623 K and 673 K (Figure 1B,C), a new magnetic phase, namely magnetite was also identified next to the spinel structure. The magnetite was formed in a relatively large quantity in the case of these two samples, which were 38.9 wt% and 48.3 wt%, respectively. The characteristic reflections, which belong to Fe_3_O_4_, were visible at 18.2° (111), 30.3° (220), 35.7° (311), 43.2° (400), 53.5° (422), 57.1° (511), and 62.3° (440) two theta degrees (PDF 19-629). At a high temperature (773 K), other phases also formed (Figure 1D), and these were the hematite and bixbyite. The latter is a manganese iron oxide (Mn,Fe)_2_O_3_ whose appearance is known to be temperature dependent [39]. The bixbyite content of the sample (773 K) was low (3.9 wt%), its reflection peaks were found at 23.3° (211), 33.0° (222), 35.8° (321), 38.4° (400), 45.3° (332), 49.5° (431), 55.2° (440), 60.5° (611), 63.9° (541), and 65.9° (622) two theta degrees (PDF 89-4836). The reflections at 24.1° (012), 33.7° (104), 35.5° (110), 40.8° (113), 49.9° (024), 54.3° (116), 62.3° (214), and 64.2° (300) correspond to the Fe_2_O_3_ phase (PDF 33-0664).

Based on the XRD analysis, we can conclude that the sample that was heat-treated at 573 K contained the spinel (MnFe_2_O_4_) phase only, while others have inhomogeneous magnetic properties (e.g., hematite). Homogeneous magnetic properties are preferred in DNA isolation since DNA adsorbed onto the surface of weakly magnetic or non-magnetic particles, such as hematite, can cause the loss of a significant amount of the tested genetic material during the purification steps.

Based on the XRD results, the crystallite sizes were calculated using the full width at the half maximum (FWHM) (Table 1). The average crystallite sizes of the manganese ferrite and magnetite particles are very similar (11–14 nm). The crystallite sizes of the manganese ferrite samples, which were synthesized by co-precipitation, vary between 5–35 nm [7,38,40,41]. In the case of the MnFe_2_O_4_ NPs, which were synthesized by a thermal treatment method followed by calcination at various temperatures from 723 to 873 K, increasing crystallite sizes between 15–23 nm were shown [42]. All in all, the size of MnFe_2_O_4_ crystallites produced in this work is in line with the literature values.

### 2.2. Transmission Electron Microscopy (TEM) Results

The smaller the nanoparticle, the larger the specific surface area for interaction with the DNA that is available, as is the amount of DNA that can be adsorbed [11]. Thus, the size distribution of the MNPs is a crucial parameter for DNA extraction applications, and this was determined by transmission electron microscopy (TEM) images taken from each magnetic nanopowder sample (presented in Figure 2A–D and shown in Appendix A Appendix A with higher resolution). These images were then analyzed using Image J software to characterize the nanoparticles. The statistical description of the nanoparticles is tabulated in Table 2 and the corresponding size distribution is shown in Figure 2E (box plot representation of these data is also available in Appendix A Appendix A).

The mean of the nanoparticle size increased slightly as the temperature increased. According to ANOVA, the samples synthesized at 573 K, 623 K, and 673 K could not be considered significantly different at a 95% confidence level. The mean particle sizes were between 11.2 and 12.6 nm and were quite close to the corresponding median values, which is characteristic of normal distribution, an important assumption for ANOVA. On the other hand, the nanoparticles prepared at the highest investigated temperature (at 723 K) yielded a statistically different particle size distribution from the three aforementioned samples. These NPs were significantly larger and had an average size of 18.4 ± 5.9 nm and the median size was 17.5 nm. Despite these two particle size clusters, all of the produced nanoparticles were sufficiently small enough to retain superparamagnetic properties [43]. It is also important to mention that since the average particle sizes obtained from TEM images are within the respective standard error of the average crystallite sizes for each of the nanoparticles obtained from the XRD results, one can consider these nanoparticles to be of a single domain. This was confirmed with a HRTEM picture, where the ordered crystal planes are clearly visible (Appendix A Appendix A). Li et al. [44] also reported preparation of cobalt ferrite nanoparticles for which the crystalline size (obtained by XRD) and the particle diameter (by TEM) are in good agreement. Based on this observation, they suggested that the vast majority of the particles are a single crystal.

The heating of the manganese ferrite particles caused them to sinter and form larger sized nanoparticles. This can be explained by the fact that the driving force of sintering is the minimization of the free surface energy caused by the elimination of interfaces, grain boundaries, and defects [45,46].

### 2.3. FTIR Measurements

As seen on the FTIR spectrum of the sample produced at 573 K, two characteristic peaks were shown, which are the absorption bands of the Mn–O and Fe–O bonds appearing at 448 cm^−1^ and 568 cm^−1^, respectively. These bands correspond to intrinsic stretching vibrations of the metal–oxygen bonds at the octahedral (448 cm^−1^) and tetrahedral (568 cm^−1^) sites for spinel MnFe_2_O_4_ [47]. The band at 1099 cm^−1^ originates from the stretching vibration of the C-O bond [48]. Other carbon-containing bonds were also identified, such as the FTIR band at 1557 cm^−1^ of the C=C bonds, whereas the two bands at 2864 cm^−1^ and 2927 cm^−1^ wavenumbers correspond to the symmetric and asymmetric stretching vibration of the aliphatic and aromatic C-H bonds, respectively [48]. Since polyethylene glycol (PEG) served as the reaction medium during MNP production, its imperfect burning could be the reason why traces of carbon were observed in the MNP samples. By increasing the heating temperature, the intensity of the νC-O, νC=C, and νC-H bands decreased. This reduction of the carbon content is a consequence of thermal oxidization and evaporation [49]. The band at 1410 cm^−1^ can be assigned to the bending vibration of the hydroxyl groups, while the stretching mode of OH located at 3429 cm^−1^ is associated with a wide band [7,50]. The -OH bands at 1410 cm^−1^ origin from the PEG, but this is found only in the samples which are made at lower synthesis temperatures. In the case of the samples synthesized at higher temperatures (673 K and 773 K), the abovementioned bands are not visible due the decomposition of the polyol. Bands were observed at 1641 cm^−1^ in all four samples (Figure 3A), which were assigned to the bending vibrations of the adsorbed water molecules [51,52] and the surface hydroxyl groups on the ferrite [53]. After the heat treatment, vaporized water molecules adsorb onto the surface of the manganese ferrite samples as they cool to room temperature. An interesting phenomenon was noted on the spectra of the samples produced at 673 K and 773 K. On the band of the metal-oxygen vibrations (568 cm^−1^), a shoulder was observed at 642 cm^−1^, which is attributed to the formation of magnetite in addition to manganese ferrite. A further increase in temperature caused the appearance of a second shoulder at 726 cm^−1^, which can be attributed to the appearance of hematite and bixbyite oxides.

The importance of hydroxyl groups on the surface of nanoparticles lies in the fact that they promote the formation of hydrogen bonds between the nanoparticles and DNA [54]. Such interaction sites enable the DNA to bind reversibly to the MNPs during the extraction process, and thus, by changing the buffer medium, it can be easily separated from the other unwanted macromolecules [55].

### 2.4. Electrokinetic Potential and Surface Area Measurements

Due to the deprotonation of surface hydroxyl groups, the electrokinetic potential showed negative average values between −11 mV and −17 mV (Figure 3B). With the increasing temperature of the heat treatment, the absolute values of the average zeta potentials showed a slight decrease. The nanoparticles electrostatically repel each other in aqueous media due to their negative surface charge, while their hydrophilic nature allows them to adequately disperse in the aqueous phase and enables them to interact with hydrophilic DNA molecules.

The MnFe_2_O_4_ sample created at 573 K contained only manganese ferrite nanoparticles, which means they were free from non-magnetic oxides. This is important because non-magnetic oxides would impair the efficiency of magnetic separation. Furthermore, the average zeta potential for this ferrite sample was the most negative (−17 ± 6 mV). Due to these properties, the manganese ferrite sample prepared at 573 K was chosen for further DNA purification tests. Surface area measurements were also performed on these samples by CO_2_ adsorption–desorption experiments at 273 K, using the Dubinin–Astakhov method. The specific surface area of MnFe_2_O_4_ prepared at 573 K was 76.0 m^2^/g. Based on the aforementioned characteristics of this nanoparticle, the MnFe_2_O_4_ sample produced at 573 K was considered promising for further biological applications [56,57,58].

### 2.5. Thermogravimetric Analysis of the Ferrite Samples

Thermal analysis studies of ferrite nanoparticles were carried out and the thermogravimetry (TG) curves of the samples are shown on Figure 4. The TG and DTG curves showed three weight losses.

As can be seen, the two samples prepared at high temperature (MnFe_2_O_4_ (673 K) and MnFe_2_O_4_ (773 K) had only minor change in mass in the temperature range studied. The evaporation of water is mainly responsible for this change which already occurred below 473 K [59,60]. These samples contained roughly 0.9 and 1.7 wt% adsorbed water as shown in Table 3. As has also been shown, there was only a small amount of organic compounds remaining in these samples.

Beside the water loss, there was a significant second weight loss in the case of the MnFe_2_O_4_ (573 K) and MnFe_2_O_4_ (623 K) samples between 500 and 600 K. This can be assigned to the decomposition of the physisorbed organic molecules which remained from ferrite synthesis, probably polyethylene glycol as was also suggested by Mukhopadhyay [48]. This finding is also in line with the TGA measurement of pure PEG reported by Massoumi et al., since PEG decomposition occurred as one step around 543–643 K [61]. Due to this decomposition, an endothermic peak appeared in the TDA curve (Appendix A Appendix A) [48] and 3.3–6.9 wt% of the samples was lost in this way. At a higher temperature (between 600 and 705 K), the oxidation of the remaining organic compounds (including the remaining PEG) occurred. This process is responsible for roughly 10 and 20 wt% of the weight loss. These organic compounds remained from the incomplete burning of the PEG during the preparation of the nanoparticle [52] due to the lower preparation temperature. The burning of the remaining organic compound (PEG) and carbon led to the appearance of an exothermic peak on the DTA curve (Appendix A Appendix A). Finally, small weight change occurs between 860 and 1016 K for each sample, ascribed to the phase transition (formation of oxides) from the ferrite [62,63,64].

### 2.6. Magnetization Measurements

The magnetization curve of the MnFe_2_O_4_ MNPs (573 K) was measured at 303 K for a magnetic field of 15,000 Oe using a vibrating sample magnetometer (VSM). The magnetic saturation (Ms) reached 72 emu/g as shown in Figure 5A. A similar Ms value (73 emu/g) was measured by Pradhan et al. in the case of annealed MnFe_2_O_4_ particles (created by co-precipitation) [41], and comparable Ms values (69.5 emu/g and 74 emu/g) have been reported in other literature as well [38,65].

The magnetization curve shows a very small hysteresis loop with low coercivity (Hc) and low remanent magnetization (Mr) as can be seen in the inlet of Figure 5A. The values of Hc (0.7 Oe) and Mr (0.1 emu/g) are quite small, indicating the superparamagnetic nature of the sample at room temperature (Figure 5B) [66]. This also supports our observation that our MnFe_2_O_4_ nanoparticles are single crystalline with a single magnetic domain [67]. Such a feature is very useful for our targeted DNA isolation application since superparamagnetic nanoparticles can be easily collected by magnets while the decantation of the supernatant is carried out. Without the presence of an external magnetic field, the magnetic properties of the nanoparticles become rather small and therefore no aggregation can occur due to the remanent magnetization of the nanoparticle. Preparation of ferrite nanoparticles with such superparamagnetic properties are not always the case. Properties of ferrite nanoparticles highly depend on the synthesis methods [44,64,68,69,70,71,72,73,74,75,76,77,78,79,80,81] (Appendix A Appendix A). 

### 2.7. DNA Binding Experiments

The DNA binding property of MnFe_2_O_4_ nanoparticles was tested using the protocol described in Section 4.6. Reversible DNA–MNP binding experiments (Figure 6A,B) were performed at least three times to verify the reproducibility of the extraction process whereby a DNA-free sample (ultrapure water used instead of cell lysate) was used as a negative control (Figure 6B). This control served to demonstrate that MnFe_2_O_4_ NPs alone cannot bind the fluorescent dye used for the visualization of agarose gels in the electrophoresis experiments. Therefore, the fluorescence signal must only come from the DNA isolated with the MNPs. As seen in Figure 6A, the DNA fluorescence bands appeared only in the elution fractions (column 2 and 3) that correspond to the purified pBAD24 plasmid, and no fluorescent signal can be seen in the supernatant fraction (column 1). The lack of a signal in the supernatant sample also means that the amount of MNPs (20 mg/mL) used was sufficient for the extraction to bind all of the the pDNA present in 5 mL of cell lysate. The second column in Figure 5A is the first fraction eluted from MNPs with 80 µL of elution buffer. To maximize the amount of pDNA extracted by the MNPs, the elution step was repeated once with the same buffer volume (Figure 6A, column 3). A smaller but still significant amount of pure DNA was extracted in the second elution step. The extra, less intense bands in the second column may be due to the different pDNA conformations [82,83]. The DNA concentration of the first elution fraction was 390.60 ± 41.55 µg/mL and 178.77 ± 16.06 µg/mL in the second fraction, as shown in Table 4. To estimate the purity of the DNA in each solution, the absorbance at 260 nm and 280 nm (A_260/280_) was measured. The typical ratio for pure DNA is in between 1.7 and 2.0 [84]. In our experiment, the A_260/280_ ratio was found to be 2.06 ± 0.03 and 2.09 ± 0.05 for the isolated pDNA in the first and second elution fractions, respectively. Although these values are slightly above the upper limit of the specified purity range, references suggest that a 260/280 absorbance ratio between 1.93 and 2.27 indicates insignificant levels of contaminants [85,86].

The DNA binding capacity of MnFe_2_O_4_ MNPs using a smaller volume of cell culture (1.5 mL instead of 5 mL) and proportionally smaller amount of MNPs (6 mg/mL instead of 20 mg/mL) was also tested (columns 1–3 in Figure 7). The 1.5 mL cell suspension volume was chosen in order to fit into the most frequently used microcentrifuge tube (Eppendorf tube), and this 1.5 mL cell suspension is the commonly suggested quantity recommended by other DNA extraction kits as well [87,88]. During the DNA extraction process, using 6 mg/mL of MNPs, a detectable amount of pDNA with good purity was obtained (Table 4). Small amount of the target pDNA remained unbonded during the DNA–MNP coincubation, since a slight DNA-coupled fluorescent band was seen in the supernatant fraction (see column 1 in Figure 7). When the used cell culture volume remained unchanged and the amount of the MNPs was increased to 20 mg/mL (columns 4–6), 1.6 times more pDNA was extracted (in terms of the total quantity of the first and second elution steps, 61.58 µg/mL and 100.73 µg/mL, respectively) compared to the previous cell culture–MNP ratio. Since pBAD24 is a low copy number plasmid, for these types of plasmids we recommend the use of an initial cell culture volume of 5 mL with a 20 mg/mL MnFe_2_O_4_ magnetic nanoparticle to yield large-scale purified pDNA with adequate purity (Table 4 and columns 7–9 in Figure 7). As shown in Table 4, when a larger initial cell suspension volume was used, the purity of the extracted product reached the upper limit of the conventional purity range of 1.7–2.0. Therefore, we can conclude that cell culture–MNP ratio of 5 mL of cell suspension to 20 mg/mL of MnFe_2_O_4_ MNPs is a good compromise for the extraction of low copy number plasmids.

## 3. Conclusions

Albeit the biocompatibility of MnFe_2_O_4_ nanoparticles being investigated previously, no relevant studies on the nucleic acid binding ability of these nanoparticles have been reported. In this work, MnFe_2_O_4_ NPs were prepared by a sonochemical combustion method. Four different temperatures were applied in their preparation (573 K, 623 K, 673 K, and 773 K). We investigated the ferrite spinel content of the particles prepared at these temperatures and found that the NPs made at 573 K were entirely in the spinel phase. XRD results and TEM images were used to characterize the crystalline size and size distribution of the metal oxide nanoparticles, respectively. The average particle sizes of all four investigated particles (573 K, 623 K, 673 K, and 773 K) were in the range of 10–20 nm. Even though all of the produced nanoparticles were small enough to exhibit superparamagnetic properties, the MnFe_2_O_4_ NPs produced at 573 K had the smallest average particle size and a sufficiently high degree of magnetic saturation. Furthermore, this sample also had the most intensive -OH band. The presence of hydroxyl groups contributes to the negative zeta potential, which also improves the colloidal stability for MnFe_2_O_4_ NPs (573 K). These features demonstrated that the MnFe_2_O_4_ (573 K) NPs were suitable for nucleic acid isolation.

In this work, we successfully extracted DNA from a complex cell lysate using MnFe_2_O_4_ MNPs. We have also demonstrated that superparamagnetic manganese ferrite nanoparticles reversibly bind pDNA according to the salt concentration of the used buffers. The amount of magnetic nanoparticles required for the isolation depends on the volume of the cell suspension. For low copy number type plasmids, a 5 mL cell suspension and 600 µL of a 20 mg/mL MNP solution was sufficient for the extraction of a large quantity of pDNA with no DNA loss after a DNA–MNP coincubation step. Both the DNA concentration measurements and the agarose gel electrophoresis images confirmed that the resulting pDNA extract was suitable for further molecular biological applications. 

We present a new biological application for MnFe_2_O_4_ MNPs and a fast and simple method for their preparation. The implemented method provides mass production of small-sized MNPs with a high DNA binding capacity. The significance of our results is demonstrated by the fact that the performance of MnFe_2_O_4_ NPs prepared at 573 K in the purification of pDNA matches that of commercially available kits.

## 4. Materials and Methods

### 4.1. Materials

The manganese ferrite nanoparticles were prepared from the following ingredients: manganese (II) nitrate tetrahydrate / Mn(NO_3_)_2_ ∙ 4 H_2_O (Carl Roth GmbH, Karlsruhe, Germany), iron (III) nitrate nonahydrate / Fe(NO_3_)_3_ ∙ 9 H_2_O (VWR International, Leuven, Belgium), and polyethylene glycol (PEG 400, [MW = 400 g/mol]) from VWR International (Fontenay-sous-Bois, France) and they were used as a reducing agent and dispersion media for the metal precursors.

For the maintenance of bacterial cell cultures and isolation of plasmid DNA with MnFe_2_O_4_ nanoparticles the following chemicals were used: tryptone, yeast extract (Neogen Culture Media, Lansing, MI, USA), sodium chloride, bacteriological agar, polyethylene glycol (VWR International, Leuven, Belgium), ampicillin sodium salt (Alfa Aesar, Kandel, Germany), a plasmid purification midi kit (Qiagen, Hilden, Germany), tris hydrochloride salt, bromophenol blue sodium salt (VWR International, Solon, OH, USA), ethylenediaminetetraacetic acid disodium salt dihydrate (Sigma-Aldrich, Louis, MO, USA), 96% ethanol (VWR International, Fontenay-sous-Bois, France), Tween 20, Gel Red nucleic acid gel stain, agarose (Merck Millipore, Billerica, MA, USA), and a 1 kb DNA ladder (New England Biolabs, Ipswich, MA, USA and Thermo Fisher Scientific, Waltham, MA, USA).

### 4.2. Synthesis of Manganese Ferrite Magnetic Nanoparticles

Manganese (II) nitrate (2.00 g) and iron (III) nitrate (6.44 g) were dissolved in 20 g of polyethylene glycol. The solution of the precursors was treated for 3 min with ultrasonic irradiation using a Hielscher UIP1000 Hdt. homogenizer (1000 W, 20 kHz) with Bs4d22 ultrasonic block sonotrode (D = 22 mm). The exposure of the liquid medium to intense ultrasonic effects results in the formation of bubbles of a few micrometers in the mixture. These bubbles then burst as the pressure increases [89], releasing thermal energy locally (forming a “hot spot”) which activates the reducing agents such as polyethylene glycol (PEG). This provides energy for the formation of highly dispersive metal hydroxide nanoparticles from the metal precursors. PEG can be removed from the dispersion produced in this way by thermal oxidation (burning). In order to do so, the PEG-based colloid system of the iron and manganese hydroxides was heated in a furnace at four different temperatures (573 K, 623 K, 673 K, and 773 K) for 3 h. After burning the polyol content, the metal hydroxide was dehydrated which resulted in magnetizable spinel nanoparticles. 

### 4.3. Physico-Chemical Characterisation of the Nanoparticles

The size and morphology of the MnFe_2_O_4_ nanoparticles were studied by high-resolution transmission electron microscopy (HRTEM, Tecnai G2 electron microscope, 200 kV (FEI Company, Hillsboro, OR, USA). Sample preparation from aqueous dispersion was carried out by placing a drop of dispersion on a 300-mesh copper grids (Ted Pella Inc., Redding, CA, USA).

The phase identification and quantification of the different oxide forms was performed with X-ray diffraction measurements realized by Rietveld analysis. A Bruker D8 diffractometer (Cu-Kα source) in parallel beam geometry (Göbel mirror) with a Vantec detector was applied. The X-ray diffraction patterns from the manganese ferrite, magnetite, hematite, and bixbyite matched the patterns of the corresponding standards, PDF 74-2403, PDF 19-629, PDF 33-0664, and PDF 89-4836, respectively. The average crystallite size of the domains was calculated using the full width at the half maximum.

The carbon content of the ferrite samples was measured by a Vario Macro CHNS element analyzer, with phenanthrene applied as a standard (C: 93.538%, H: 5.629%, N: 0.179%, S: 0.453%) from Carlo Erba Inc. (Emmendingen, Germany). The carrier gas was helium (99.9990%), whereas oxygen (99.995%) was used as an oxidative atmosphere.

The zeta potentials of the nanoparticles were examined based on electrophoretic mobility measurements by applying laser Doppler electrophoresis using Zetasizer Nano ZS (Malvern Panalytical Ltd., Malvern, United Kingdom).

The identification of the spinel chemical bonds was carried out with Fourier transform infrared spectroscopy using a Vertex 70 spectroscope (Bruker Corporation, Billerica, MA, USA). During the preparation, a 10 mg sample was pelletized with 250 mg potassium bromide, and the measurements were realized in the transmission mode.

The specific surface area of the ferrite samples was examined by CO_2_ adsorption–desorption measurements at 273 K by using an ASAP 2020 sorptometer (Micromeritics Instrument Corporation, Norcross, GA, USA), based on the Dubinin–Astakhov method.

The measurement of moisture content, loss of volatile components, and the carbon content of the manganese ferrite samples was carried by thermogravimetric analysis (TGA) using a TG 209 Tarsus thermo-microbalance device (Erich Netzsch GmbH & Co. Holding KG, Selb, Germany). A nitrogen (4.5) and oxygen (5.0) mixture was used as an oxidative atmosphere in the measurements. The flow rates were set to 6 mL min^−1^ and 14 mL min^−1^ for the oxygen and nitrogen, respectively. The heating rate was 10 K min^−1^ in the 323–1073 K temperature range.

The magnetic characterization of ferrite nanoparticles was carried out with a vibrating sample magnetometer (Lake Shore Cryotronics Inc., Westerville, OH, USA), using the 8600 VSM system at a 303 K temperature. The magnetization (M) versus the applied magnetic field (H) was performed over H up to 15,000 Oe.

### 4.4. Biological Characterization of Nanoparticles

Nucleic acid isolation was carried out by means of a Mega Star 1.6R centrifuge (VWR International, Leuven, Belgium) and an NB-205QF cooling and shaking incubator (N-Biotek, Gyeonggi-do, Republic of Korea). The effectiveness of the nucleic acid isolation by MNPs was verified with a Mini-Sub Cell GT horizontal agarose gel electrophoresis system (Bio-Rad Laboratories, Hercules, CA, USA), a NanoDrop One Microvolume UV–Vis spectrophotometer (Thermo Fisher Scientific, Waltham, MA, USA) and a gel documentation system (Uvitec, Cambridge, United Kingdom).

### 4.5. Growth of Escherichia coli

An *Escherichia coli* DH5α bacterial strain with ampicillin antibiotic resistance on a pBAD24 plasmid was used for pDNA extraction. The cell cultures were preserved on agar plates in our laboratory at 37 °C. For preparing fresh bacterial cell suspensions, 35 mL Luria–Bertani (LB) medium containing 100 µg/mL of ampicillin was inoculated using a sterile inoculation loop. After overnight (18–20 h) incubation at 37 °C with vigorous shaking (160 rpm), the cell suspension was divided into centrifuge tubes, each containing 5 mL, as this quantity was used for each pDNA isolation experiment. The remaining suspension aliquots were stored at −20 °C for later experiments.

### 4.6. pDNA Isolation with MnFe_2_O_4_ Magnetic Nanoparticles

For the isolation process, bacterial cells were centrifuged for 5 min on 6000× *g* and the supernatant medium was decanted. Solutions P1–P3 from the plasmid purification midi kit (Qiagen) were used to lyse the cells and precipitate the macromolecules. Cell lysis was performed following the (Qiagen) manufacturer’s recommendations. With the help of the neutralization buffer, only the plasmid DNA was renatured in the solution. Using a high-speed centrifuge (14,500× *g*, 10 min) pDNA was separated from cell debris and the irreversibly denatured macromolecules. The nucleic acid binding capacity of the tested MnFe_2_O_4_ nanoparticles was carried out in such a way that the crude extract was mixed with the nanoparticles and the subsequent isolation steps were performed using a modified version of a procedure from the literature [90]. The supernatant solution of the cell lysate (600 µL) was added to 600 µL of a 20 mg/mL MnFe_2_O_4_ dispersion suspended in binding buffer (containing 2.5 M NaCl, 1 M Tris-HCl pH 8.0, 0.5 M EDTA, 20% (*w*/*v*) PEG 6000 and 0.05% Tween 20). Eppendorf tubes were flipped upside down for 10 min, thereby creating MNP–DNA complexes. This was followed by a 5 min incubation step (at room temperature) on a strong external magnet (a magnetic stand). The emerging crude supernatant fraction (containing potentially unbound DNA molecules and contaminants) was collected and pipetted into a new microcentrifuge tube and the residual pellet (the DNA–MNP complex) was washed as follows. In total, three washing steps were carried out, each using the same 1 M Tris-HCl pH 7.5 buffer with 96% ethanol. First, 1 mL of the wash buffer was added to the suspension of DNA-coated magnetic nanoparticles. After vortexing the tubes for a few seconds, 2 min incubation was performed on the magnet. After removal of the supernatant (while the Eppendorf tubes remained on the magnet), this washing step was repeated with 30 s incubation on the magnet. During the third washing step, 500 µL of washing solution was used and after vortexing, the samples were placed on the magnet for 2 more minutes. Finally, the supernatant was removed with a pipette. A short centrifugation step was carried out and the excess wash buffer was removed. The tubes were dried for 15 min in the 37 °C incubator with an open cap. The samples were eluted with Tris-HCl (pH 8.5). Within these circumstances, DNA dissociated from the MNPs. Eighty µL of the elution buffer completely wet the DNA–MNP complex, and after 10 min incubation at 37 °C, the purified pDNA was separated from the MNPs using 5 min incubation on a magnetic stand. To maximize the amount of extracted pDNA, an additional elution with a 80 µL volume was performed (Figure 8). 

### 4.7. Gel Electrophoresis

To confirm the success of reversible DNA–MNP binding, gel electrophoresis experiments were carried out using 0.75-cm thick 1.0 *w*/*v*% agarose gels (1 g agarose powder, 100 mL of Tris-Acetate-EDTA buffer (TAE; 40 mM Tris-base, 20 mM acetic acid, 1 mM EDTA [91]). The running buffer for electrophoresis was also the TAE buffer. An amount of 6× gel loading dye solution was used (30 *v*/*v*% glycerol, 0.25 *w*/*v*% bromophenol blue dye, and ultrapure water) to provide the requisite density for loading the sample into the well and to monitor the progress of electrophoresis [91]. Using a 6× concentrated loading buffer means that the mixture prepared for electrophoresis contained one part DNA loading dye and five parts isolated DNA sample. The electrophoresis was run at 90 V for 45 min.

### 4.8. Determination of pDNA Concentration

DNA concentration measurements were performed with microvolume nucleic acid quantification (UV–Vis NanoDrop (Thermo Fisher Scientific, Waltham, MA, USA)) using the absorbance measured at 260 nm. To determine the purity of the extracted pDNA, the absorbance of the sample at 280 nm was also measured. A typical ratio of the absorbance at 260 nm and 280 nm (A_260/280_) for a pure DNA solution is considered to be between 1.7 and 2.0.

## Figures and Tables

**Figure 1 ijms-24-02156-f001:**
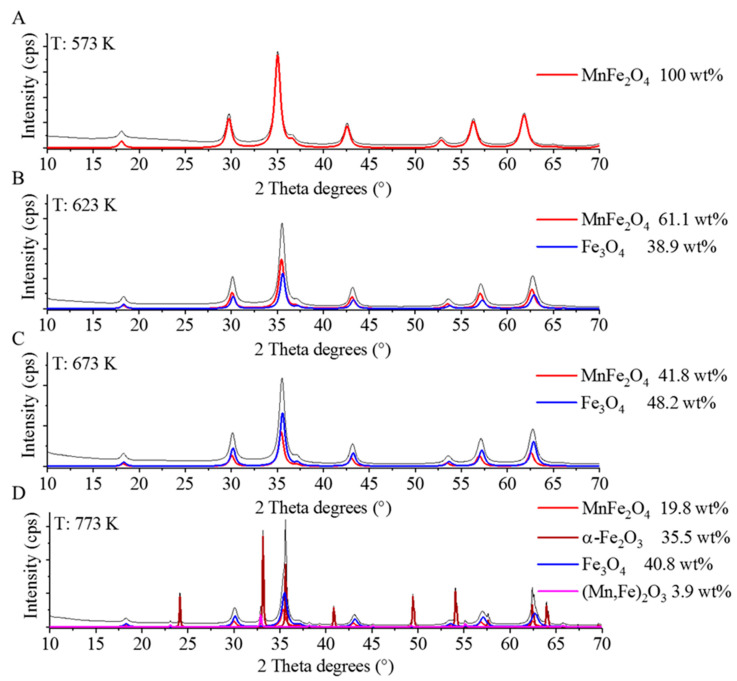
X-ray diffractograms of the ferrite samples produced at 573 K (**A**), 623 K (**B**), 673 K (**C**) and 773 K (**D**) temperatures.

**Figure 2 ijms-24-02156-f002:**
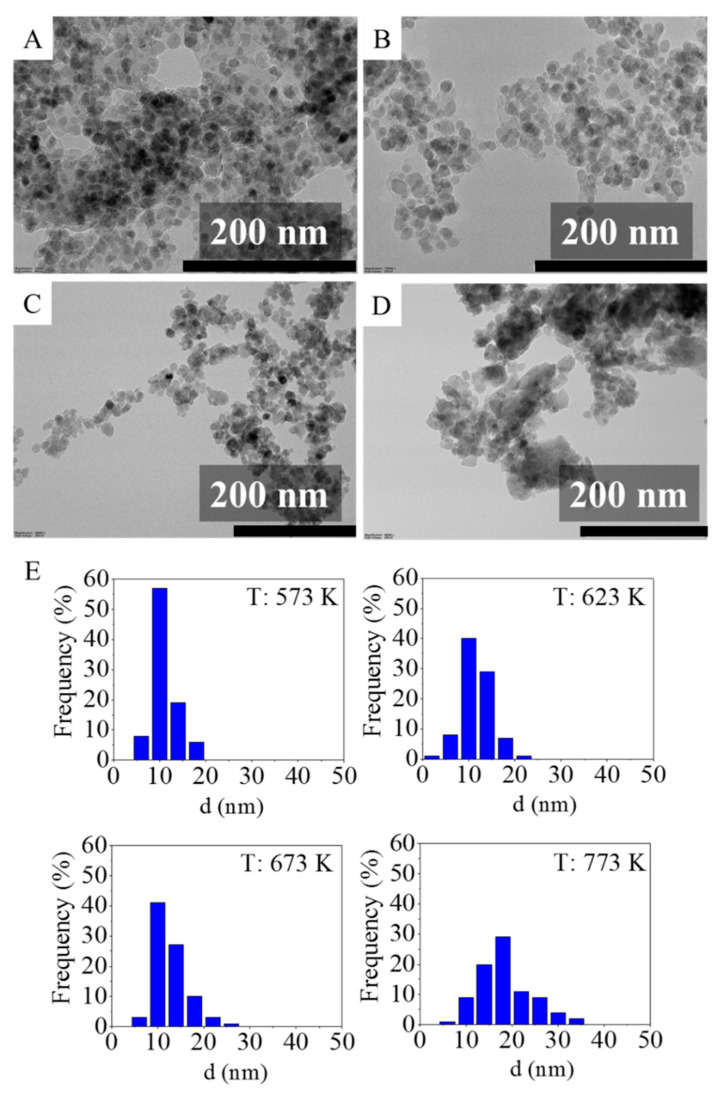
TEM images of the nanoparticles synthesized at 573 K (**A**), 623 K (**B**), 673 K (**C**), and 773 K (**D**), and their size distribution analysis on histograms (**E**).

**Figure 3 ijms-24-02156-f003:**
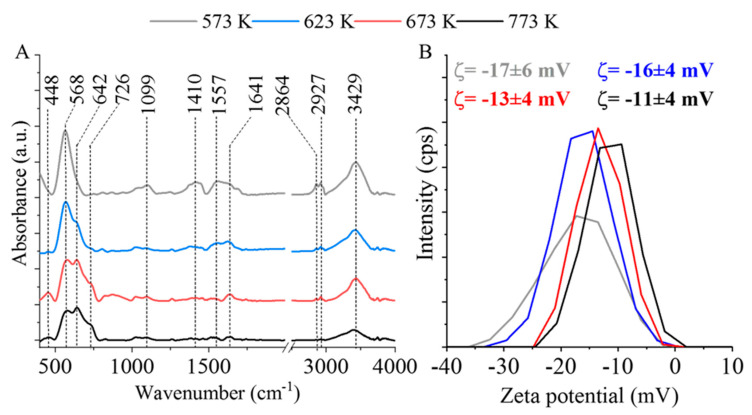
FTIR spectrum (**A**) and zeta potential distribution (**B**) of the nanoparticle samples produced at four different temperatures.

**Figure 4 ijms-24-02156-f004:**
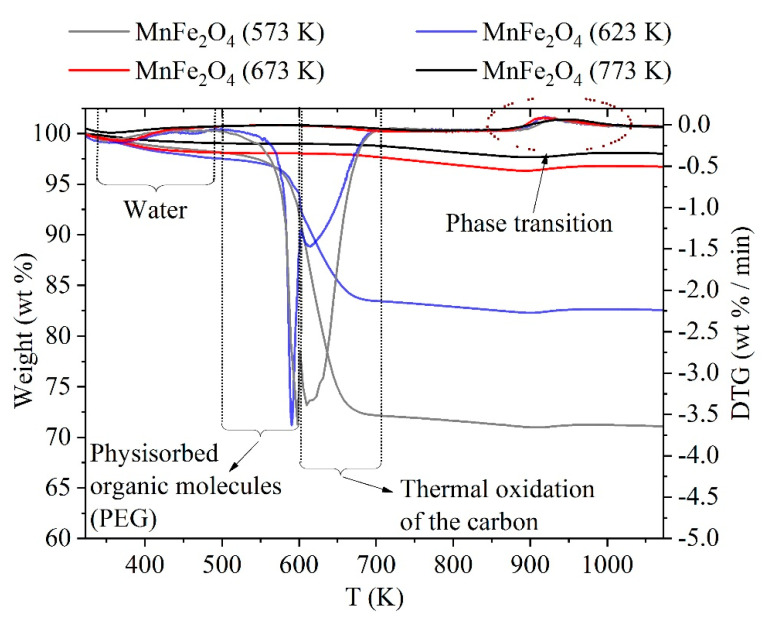
TG and DTG curves of the ferrite samples in air atmosphere.

**Figure 5 ijms-24-02156-f005:**
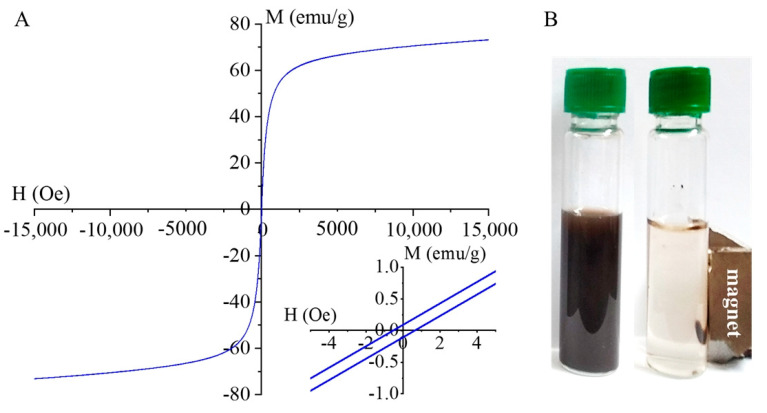
Magnetization curve of MnFe_2_O_4_ (573 K) (**A**) and its magnetic separability by using a magnetic field (**B**).

**Figure 6 ijms-24-02156-f006:**
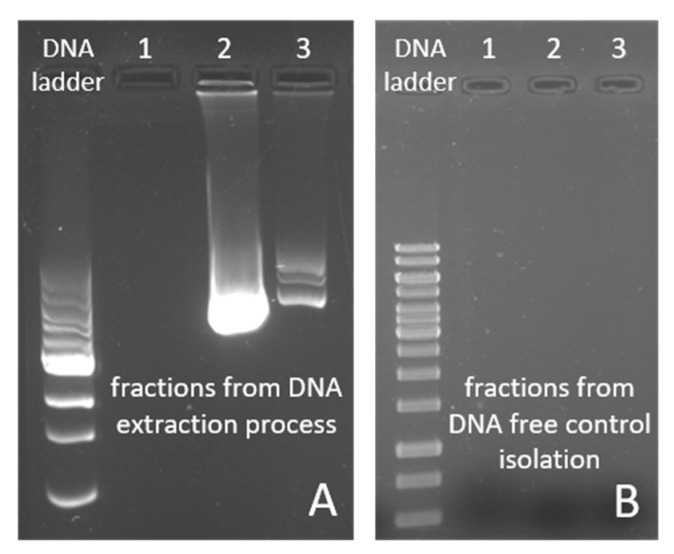
Agarose gel electrophoresis image of isolated pBAD24 plasmid DNA extracted with MnFe_2_O_4_ MNPs (**A**) and DNA-free control isolation fractions (**B**). In both pictures, the DNA ladder notation refers to a 1 kb DNA marker. (1) is the supernatant fraction, (2) marks the first elution fractions using 80 µL of elution buffer, and (3) indicates the second elution fractions (also eluting with 80 µL of buffer).

**Figure 7 ijms-24-02156-f007:**
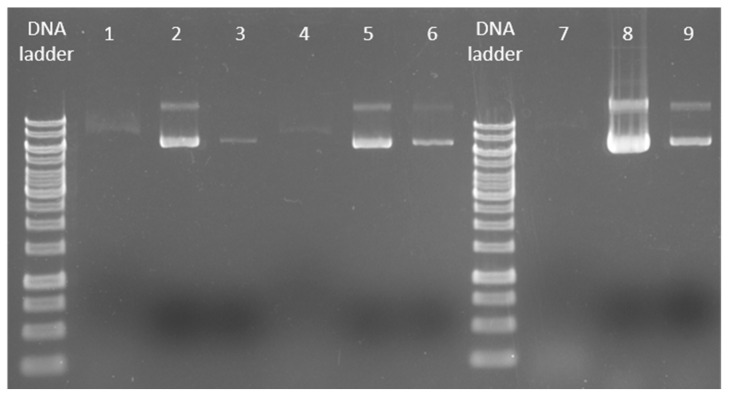
DNA binding capacity of the MnFe_2_O_4_ MNPs.

**Figure 8 ijms-24-02156-f008:**
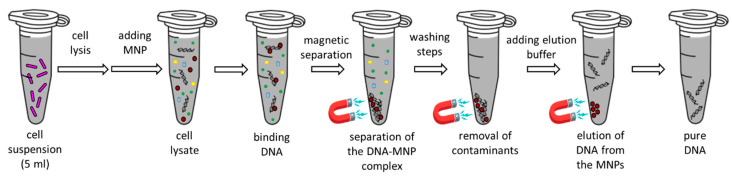
pDNA isolation with MnFe_2_O_4_ magnetic nanoparticles.

**Table 1 ijms-24-02156-t001:** Average crystallite sizes of the nanoparticles obtained from the XRD results.

T (K)	MnFe_2_O_4_ (nm)	Fe_3_O_4_ (nm)
573	11	-
623	11	14
673	13	12
773	14	14

**Table 2 ijms-24-02156-t002:** Results of the particle size distribution analysis based on TEM images.

T (K)	Mean (nm)	Min. (nm)	Max. (nm)	Median (nm)	P90 (nm)	P95 (nm)	P99 (nm)
**573**	11.2 ± 2.6	5.6	18.6	10.7	14.5	17.1	18.6
**623**	11.7 ± 3.2	3.4	20.9	11.3	15.9	16.4	20.9
**673**	12.6 ± 3.8	5.9	25.3	11.9	17.8	19.2	25.3
**773**	18.4 ± 5.9	4.2	34.1	17.5	27.5	29.5	34.1

**Table 3 ijms-24-02156-t003:** The weight losses of the ferrite samples.

	Adsorbed Water (wt%)	Physisorbed Organic Compounds (wt%)	Remaining Carbon Forms (wt%)
**MnFe_2_O_4_ (573 K)**	2.5	6.9	19.25
**MnFe_2_O_4_ (623 K)**	1.9	3.3	10.9
**MnFe_2_O_4_ (673 K)**	1.7	0.24	1.0
**MnFe_2_O_4_ (773 K)**	0.9	0.15	0.11

**Table 4 ijms-24-02156-t004:** DNA concentration of the MnFe_2_O_4_-purified pDNA samples.

Volume of Initial Cell Suspension (mL)	Concentration of MnFe_2_O_4_ MNP (mg/mL)	Concentration of Purified pDNA in First (80 µL) Elution (µg/mL)	A_260/280_	Concentration of Purified pDNA in Second (80 µL) Elution (µg/mL)	A_260/280_
**1.5**	6	44.95 ± 19.87	1.87 ± 0.12	16.63 ± 12.76	2.04 ± 0.14
**1.5**	20	63.8 ± 13.01	1.87 ± 0.07	36.93 ± 8.94	1.99 ± 0.07
**5**	20	390.60 ± 41.55	2.06 ± 0.03	178.77 ± 16.06	2.09 ± 0.05

## Data Availability

The data presented in this study are available on request from the corresponding author. The data are not publicly available due to the policy of the University of Miskolc.

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
