# Peer review of "A Simplified and Efficient Method for Production of Manganese Ferrite Magnetic Nanoparticles and Their Application in DNA Isolation"

_ijms, 2023, doi:10.3390/ijms24032156_

Round 1

Reviewer 1 Report

Report on the revised manuscript "Sonochemical assisted combustion synthesis of manganese ferrite magnetic nanoparticles and their application in DNA isolation" (Ms. No.: ijms-1957204)

This paper describes the use of manganese ferrite (MnFe2O4) nanoparticles, prepared through ultrasonic cavitation,  for the reversible binding of DNA chains. Unfortunately, the manuscript presents many deficiencies that prevent the recommendation for its acceptance in the present form.

Below, I will list some of these problems identified:

1.    The Abstract needs to be rewritten to eliminate problems such as a)“the DNA desorbs from the MNPs by decreasing the ion concentration of the elution buffer” [Please explain better which ions are these], and b) “the electrophoresis confirmed that high amount of pDNA with adequate purity” [Please indicate the origin of the pDNA].

2.    The paper needs to be better organized in terms of its different sections. Several key concepts and information are listed in the text before being properly defined or adequately discussed.

3.    The Introduction section needs to be improved, as there is an extensive literature concerning the use of magnetic extraction of DNA and pDNA, not adequately reviewed. See, for instance for instance,  10.1039/c9ay02690h; 10.3390/mi11030302; https://doi.org/10.1016/j.aca.2021.338762, and 10.1016/j.ab.2019.03.013, and references therein.

4.    There are many examples of lack of precision in the text that sometimes makes for a hard understanding at first reading. Here are just a few examples: a) “The main reason for their wide-range applicability is their ferrimagnetic property which shows particle size dependence due to the random orientation of their magnetic domains.” [Please rewrite]; b) “The amount of the absorbed DNA depends on the accessible surface area of the adsorbent, which can be achieved by formation of small particle sizes [9], thus it was important to determinate [sic] the size distribution of the MNPs. [????Please rewrite.]; c) “The presence of carbon could be formed [???] due to imperfect burning”.

5.    For the benefit of the general reader, the concept of pDNA should be properly explained.

6.    Please explain the effect of magnetization upon DNA binding capabilities for a better understanding of the statement “Homogeneous magnetic properties are relevant in DNA isolation since adsorbed DNA on the surface of weakly magnetic particles can cause loss of significant amount of the tested genetic material”.

7.    The speculative tone sometimes adopted (“which could be a sign of formation of a new oxide form (Fe3O4) beside MnFe2O4”, “The possible significance of the presence of hydroxyl groups is”) should be avoided.

8.    The authors should elaborate further on the results that show an A260/280 ratio of the order of (2.06 ± 0.03) μg/mL and (2.09 ± 0.05 μg/mL) for the isolated pDNA since a ratio of ~1.8 is generally accepted as an indication of a “pure” DNA fraction. A ratio of ~2.0 is generally accepted as “pure” for RNA.

For the above reasons, I would recommend that the manuscript should be submitted for a minor revision.

Reviewer 2 Report

Please check the attached file

Reviewer 3 Report

The manuscript entitled "Sonochemical assisted combustion synthesis of manganese ferrite magnetic nanoparticles and their application in DNA isolation" publishes the results of a study on the use of manganese ferrite nanoparticles for DNA isolation by reversible binding. This study has a novelty and is of scientific interest. The article is written consistently, logically, the text is clear, the information is easy to perceive. The article contains a sufficient number of references to literary sources, which are given correctly and in place. It is proposed to accept the article after minor changes.

1) Line 56. The authors could add that studies of composite materials consisting of magnetic nanoparticles and DNA have already been conducted, for example, https://doi.org/10.3390/polym14020344

2) Lines 102-107. For the particle size distribution of the samples, another method would be more suitable – measuring the hydrodynamic size using Doppler velocimetry on a Zetasizer-type device.

3) Lines 242-245. In conclusion, it is not desirable to make a direct description of the results of the experiment, it is better to leave only the consequences and generalized conclusions. This improves the perception of the material.

4) Also in conclusion, at the end, you can add one or two sentences devoted to the description of possible prospects for using the method used.

Round 2

Reviewer 2 Report

The authors did not respond to most of the observations. An extensive review with resubmission was requested. They did a cursory review of the work.

1. The degree of novelty does not emerge from the work. The authors reason that they have not worked in this field before, a simple search shows at least 10 articles in the field of nucleic acids fished with the help of MnFe2O4. Consequently, the question was not answered.

What is the purpose of the study? There are many similar studies in the literature. Nowhere in the paper did I see the degree of novelty, of interest for the readers of the IJMS journal, being highlighted

https://www.ncbi.nlm.nih.gov/pmc/articles/PMC7699708/

https://www.sciencedirect.com/science/article/pii/S0010854522004040

https://www.mdpi.com/2079-4991/10/11/2297

https://www.tandfonline.com/doi/full/10.1080/21691401.2018.1523182

https://pubs.acs.org/doi/10.1021/acsomega.0c05382

https://www.sciencedirect.com/science/article/pii/S2468519417300836?via%3Dihub

https://pubs.rsc.org/en/content/articlelanding/2014/NJ/C4NJ01182A

http://repository.bose.res.in:8080/jspui/bitstream/123456789/1823/1/Design%20and%20development%20of%20bioactive%20%2B%C2%A6-hydroxy%20carboxylate%20group.pdf

https://assets.researchsquare.com/files/rs-1931557/v1/e3e9c063-8246-4a9e-87c7-5b9a39ba1b53.pdf?c=1661579965

2. Also, suggested to include the recent references in the introduction part. Recent articles are understood from the last 2 years.

3. Compare XRD results with other articles. Compare with several articles in the field and switch to a table. What do you bring in addition to literature? At the moment it doesn't work.

4. Boxplots are not relevant, make the particle distribution histogram.

5. The variation of crystallite size and particle size with respect to temperature is not explained. There is information in the literature. Why don't you use? The notification was also made in the previous revision.

6. Why is the size of the particles larger than the size of the crystallites? There is information in the literature. Why don't you use? The notification was also made in the previous revision.

7. In discussions about structural analysis, XPS measurements are missing. Let these measurements be added to have a substrate for the correlation of the discussions. The notification was also made in the previous revision.

8. It is very difficult to follow the scale in the TEM images. To increase the writing, to unify the writing in all figures and graphs. The notification was also made in the previous revision.

9. If the TEM results are poor, why do AFM? There you get additional information besides particle size, roughness, height, etc. The notification was also made in the previous revision.

10. In the FT-IR part, the reference sources are still missing. How did the authors interpret the vibration bands when there are no bibliographic sources or international databases for most of them. What did they report to?

11. What kind of organic compounds do you have at 573, 623 and 673 K? Do a thermal analysis (TG-DTA) and see at what temperature the thermal decomposition of organic compounds takes place. On what scientific basis, without the study of thermal analysis, do you make the assumption that that organic compound did not decompose at temperatures of 573, 623, 673 K?  To complete the work with thermal analysis.

12. What vibration of the hydroxy bond are you talking about at 1410 and 3429 cm-1? do you have alcohol at high temperatures? Hydroxyl groups are characteristic of inorganic bases, alcohols and phenols, inorganic compounds and water. It is not clear to which class you are assigned.

13. Where are the magnetic measurements at the other 3 temperatures? If you made XRD, IR, TEM measurements at 4 temperatures, why don't you also make 4 VSM measurements? What do you compare a single sample with?

14. To detail the interpretation of Magnetic measurements by comparison with literature studies.

Compare saturation magnetization and coercivity with other ferrites in the literature.

15. References are not written according to the guide (not all authors are listed in all references, they are not marked with initials, the titles of the articles and journals are not mentioned in many places, the volumes or pages are missing).

16. I recommend improving the quality of the English language.

Round 3

Reviewer 2 Report

Compared to the previous revision, the work was improved. There are still unresolved aspects in the work following the first 2 revisions:

1. A series of articles that deal with the theme of the paper are not cited in the paper. I have given some examples below. I believe that in a scientific work it is necessary to present very transparently what was achieved before.

10.1039/d1ra03216j

https://doi.org/10.3390/ijms232214145

https://doi.org/10.1186/s12951-020-00613-6

https://doi.org/10.1021/la047206o

10.1088/2053-1591/aa5d93

https://doi.org/10.3389/fchem.2021.629054

https://doi.org/10.1039/D0NA00967A

2. Table 1 from the additional material should be included in the work and you should enter your values ​​in the table to make it easier to compare your results with the literature.

3. It is impossible for the size of the crystalline phases in XRD to be larger than the size of the particles formed by the crystalline and amorphous phases recorded in TEM. Do an AFM, XPS analysis.

4. In discussions about structural analysis, XPS measurements are missing. Let these measurements be added to have a substrate for the correlation of the discussions. The notification was also made in the previous 2 revision. An extension of time for the measurements can be requested, and the value of the work would increase considerably.

5. In order to clearly see the distribution of the elements in the analyzed samples, elemental analysis, XPS and EDX should be performed.

6. If the TEM results are poor, why do AFM? There you get additional information besides particle size, roughness, height, etc. The notification was also made in the previous 2 revision.

7. TEM histograms lack error bars.

8. At temperatures of 573 K and 623 K, is there still an organic component (alcoholic OH groups)? Can you give some bibliographic sources that confirm this hypothesis?

9. In the part of the thermal analysis, you state that in the range of 600-700 oC the thermal oxidation of carbon takes place, and it is the last chemical process. When the formation of MnFe2O4 takes place. It is not clear from the TG-DTG diagrams. It would be useful to add DTA so that you can discuss the exothermicity and endothermicity of the processes.

10. For the readers of the IJMS journal, it would be easier to follow the paper if the useful information from the supplementary material was in the paper. It is not right to discuss in the text of the paper about figures from the supplementary material, in the context where there is no page limit at IJMS.

11. Where are the magnetic measurements at the other 3 temperatures? If you made XRD, IR, TEM measurements at 4 temperatures, why don't you also make 4 VSM measurements? No conclusions can be made if the magnetic measurements for all the samples discussed in the paper are not presented. From where you can draw the conclusion that these samples have the best magnetic properties necessary for the chosen applicability if you do not have VSM measurements for the other samples.

12. For the third time, I make the observation that the references should be in the format of the IJMS journal.

- Why do you write the magazine title in italics and not the magazine title for each reference?

- Why don't you abbreviate the journal for each reference?

- Only the volume is requested to go to the references and not the issue. Why do you write the volume in bold when the instructions ask for italics?

- Why don't you write the year in bold as requested in the instructions?

- the page must be entered only with a number without "p"

13. I recommend improving significative the quality of the English language.

Author Response

See below
